# Double-stranded RNA virus outer shell assembly by *bona fide* domain-swapping

Zhaoyang Sun[1,*], Kamel El Omari[1,*], Xiaoyu Sun[2], Serban L. Ilca[1], Abhay Kotecha[1], David I. Stuart[1], Minna M. Poranen[2] & Juha T. Huiskonen[1,2]

Correct outer protein shell assembly is a prerequisite for virion infectivity in many multi-shelled dsRNA viruses. In the prototypic dsRNA bacteriophage φ6, the assembly reaction is promoted by calcium ions but its biomechanics remain poorly understood. Here, we describe the near-atomic resolution structure of the φ6 double-shelled particle. The outer $T = 13$ shell protein P8 consists of two alpha-helical domains joined by a linker, which allows the trimer to adopt either a closed or an open conformation. The trimers in an open conformation swap domains with each other. Our observations allow us to propose a mechanistic model for calcium concentration regulated outer shell assembly. Furthermore, the structure provides a prime exemplar of *bona fide* domain-swapping. This leads us to extend the theory of domain-swapping from the level of monomeric subunits and multimers to closed spherical shells, and to hypothesize a mechanism by which closed protein shells may arise in evolution.

[1] Division of Structural Biology, Wellcome Trust Centre for Human Genetics, University of Oxford, Roosevelt Drive, Oxford OX3 7BN, UK. [2] Department of Biosciences, University of Helsinki, Viikinkaari 9, Helsinki 00014, Finland. * These authors contributed equally to this work. Correspondence and requests for materials should be addressed to M.M.P. (email: minna.poranen@helsinki.fi) or to J.T.H. (email: juha@strubi.ox.ac.uk).

The protein shells of icosahedral viruses assemble with remarkable accuracy from smaller subunits[1–3]. Accumulating evidence suggests that such viruses can be grouped into a fairly small number, perhaps less than a dozen, of viral lineages, each sharing a set of common assembly principles[4,5]. However, explaining how such closed, highly symmetric shells may have arisen during evolution poses a challenge, as formation of a closed shell first requires a network of complementary subunit–subunit interactions with roughly 50% of the surfaces buried from solvent[6]. One of the most complex examples of such interactions is the capsid of bacteriophage HK97 where the capsid proteins are intertwined to form a topologically linked 'chain mail' organization[7]. A simpler question to ask is how do multimeric proteins evolve from monomers? It has been suggested that this can be achieved by a mechanism called 'domain swapping'[8,9]. In the resulting multimer, the domain-swapped subunits in an 'open' conformation recapitulate the domain-domain interactions of the original 'closed' conformation of the monomer. This concept explains how mutations increasing the flexibility of a linker region of the monomer can facilitate domain swapping to give rise to the multimer.

Domain-swapping may play a role not just in the assembly and evolution of multimeric proteins, but also of viral shells[10–12]. The structure of rice yellow mottle virus (RYMV; genus *Sobemovirus*), and its comparison to the related southern cowpea mosaic virus (SCPMV), provides an interesting exemplar[12]. In SCPMV, the $T = 3$ icosahedral capsid is formed in an usual way by trimers in a compact, closed conformation[13]. In RYMV, however, the trimers are atypical and adopt an open conformation, swapping domains with the neighbouring trimers and creating long range interactions, to which the increased stability of the capsid has been attributed[12]. However, it remains unclear whether RYMV trimers can exist in both open and closed conformations.

We use bacteriophage φ6, a member of the *Cystoviridae* family, as a model system to study the assembly of a complicated multi-shelled virus. The inner $T = 1$ shell, composed of protein P1, forms a dodecahedral polymerase complex (PC) harbouring the three viral dsRNA segments (S, M, and L)[14,15], the viral polymerase P2 (refs 16,17), the hexameric packaging protein P4 (refs 18,19), and the assembly cofactor P7 (refs 20,21). Unlike eukaryotic dsRNA viruses, φ6 and other cystoviruses undergo a dramatic conformational change during RNA packaging and replication, from an empty collapsed PC to a fully expanded, packaged PC[22,23]. The expanded PC is covered by a layer of P8, forming the outer $T = 13$ shell[23]. This double-shelled particle, or nucleocapsid (NC), is itself covered by a protein-lipid envelope during virion assembly[24,25]. A remarkable aspect of the φ6 system is that PC and NC particles can be assembled *in vitro* from purified protein and RNA components[26]. In addition to φ6 bacteriophage (*Cystovirus*), many other dsRNA viruses, such as aquareovirus (*Aquareovirus*)[27], mammalian reovirus (*Orthoreovirus*)[28], blue tongue virus (*Orbivirus*)[29,30], rice dwarf virus (*Phytoreovirus*)[31,32], rotavirus (*Rotavirus*)[33], and Banna virus (*Seadornavirus*)[34], form a multi-shelled particle with an inner $T = 1$ and one or more outer $T = 13$ shells. While structures for some of these have been solved to near-atomic resolution shedding light into molecular interactions in the $T = 1$ and $T = 13$ shells[27,29,30,33], assembly of the $T = 13$ shell in φ6 NC has remained elusive.

Here we present the structure of φ6 NC to near-atomic resolution using electron cryomicroscopy and single particle analysis. The structure of the outer NC shell reveals an intricate domain-swapped architecture created by P8 trimers. The majority of the P8 subunits (540 of 600) are in an open conformation, swapping domains with their neighbours. The inter-trimer

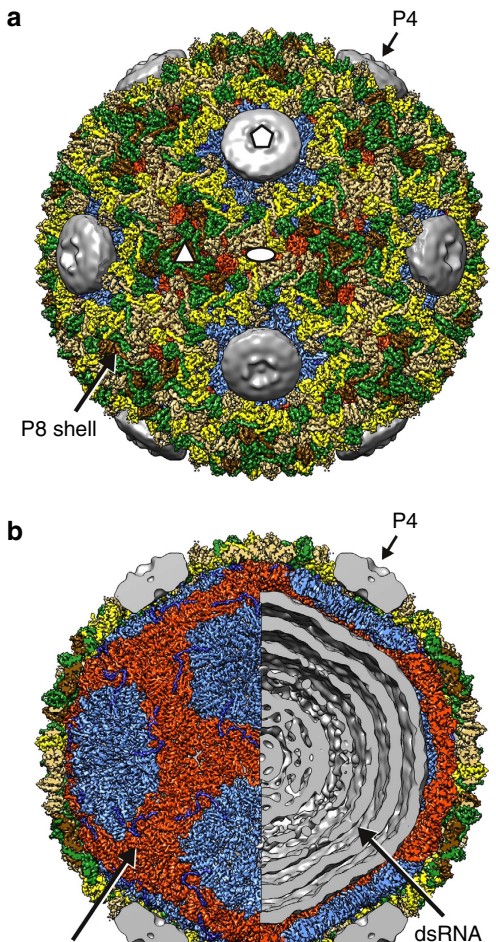

**Figure 1 | Cryo-EM structure of bacteriophage φ6 nucleocapsid.**
(**a,b**) Three-dimensional reconstruction of the NC at 4.0-Å resolution. The P8 trimers forming the outer protein shell are coloured in yellow (P8$_Q$), green (P8$_R$), gold (P8$_S$) and brown (P8$_T$). Half of the P8 density was removed to reveal in **b** the inner protein shell composed of P1$_A$–P1$_B$ asymmetric dimers, coloured in light blue (P1$_A$) and red (P1$_B$). The elongated density corresponding to the P4 C-terminus is shown in dark blue. Residual density, not assigned to the aforementioned icosahedrally ordered components was filtered to 15-Å resolution to visualize in **a,b** the symmetry-mismatched P4 hexamers at the icosahedral five-fold vertices and the dsRNA in **b**. See also Supplementary Fig. 1.

interactions recapitulate the intra-trimer interactions of the closed conformation that were also observed within the NC. These observations not only provide a possible explanation for the calcium-regulated assembly of the φ6 NC particle, but they also constitute an exemplar of *bona fide* domain swapping in a viral shell, allowing us to propose a mechanism for how such complex shells may have arisen during evolution.

## Results
**Near-atomic resolution structure of the φ6 nucleocapsid.** We determined the structure of φ6 double-shelled particle by electron cryomicroscopy (cryo-EM; Fig. 1; Table 1). A high-resolution cryo-EM data set was acquired by imaging purified φ6 NC samples, some of which had spontaneously lost the RNA genome (Supplementary Fig. 1a). Two-dimensional image classification revealed the morphology of NC particles

**Table 1 | Nucleocapsid cryo-EM data collection parameters.**

| Parameter | Value |
|---|---|
| Voltage (kV) | 300 |
| Magnification ($\times$) | 37,037 |
| Defocus ($\mu$m) | 0.3–3.0 |
| Dose rate ($e^-$ per pixel per s) | 5 |
| Frames | 22 |
| Frame length (s) | 0.2 |
| Total dose ($e^-$ per $Å^2$) | 16 |
| Micrographs | 903 |

cryo-EM, electron cryomicroscopy.

(Supplementary Fig. 1b). Some particles had lost either fully or partially the outer P8 shell (Supplementary Fig. 1a,b). Complete double-shelled particles were used to calculate a three-dimensional reconstruction using a 'gold-standard' refinement approach (see Methods; Table 1). The overall resolution of the icosahedrally symmetric areas of the reconstruction was 4.0 Å, as determined by Fourier shell correlation (FSC, threshold = 0.143; Supplementary Fig. 1c). In many areas the local resolution was approaching 3.0 Å, while some areas, especially solvent exposed loops and areas around the icosahedral five-fold vertices in the outer protein shell were resolved to a lower resolution (Supplementary Fig.1d–i). The resolution was sufficient to not only trace the course of the polypeptide chain but also to align the sequence to the structure and refine the structure, for nearly all of P1 and P8 residues (see Methods). The P4 hexamers located at the icosahedral five-fold axes of symmetry were not resolved due to the symmetry mismatch (Fig. 1). Density corresponding to the three linear dsRNA genome segments appeared as concentric layers spaced ∼3 nm apart but the detailed organization of the segments could not be addressed using icosahedral reconstruction (Fig. 1b).

**Architecture of the nucleocapsid outer shell**. The NC structure revealed, consistent with our earlier observations[23], a total of 200 P8 trimers forming the outer protein shell, following $T = 13$ laevo quasi-symmetry (Fig. 1a). Four types of trimers ($P8_Q$, $P8_R$, $P8_S$ and $P8_T$) occupy the different positions in the shell: Q-type trimers around the icosahedral five-fold axis of symmetry, S-type trimers adjacent to the two-fold axis, T-type trimers on the three-fold axis and R-type trimers between the other trimers. Each of the 60 asymmetric units of the shell thus consists of 10 P8 chains, as the $P8_T$ trimers contribute only one chain to each asymmetric unit.

The $T = 13$ shell revealed a remarkable domain swapped organization of P8 trimers (Fig. 2). The relatively high resolution of the reconstruction allowed modelling the fold of the P8 structure (residues 3–147; Fig. 2a,b; Supplementary Fig. 2). The structure is α-helical and can be divided into two domains that are connected by an elongated linker (Fig. 2c,d): an N-terminal peripheral domain (PD; residues 1–83) is followed by a 10-residue long elongated loop (residue 84–93) and a C-terminal core domain (CD; residues 94–149). The loop ends in a proline residue (Pro93), which breaks the preceding helix α6 of the CD (Supplementary Fig. 2). The CDs intertwine together to form the core of the P8 trimer. The PDs, due to the elongated nature of the linker, are located a considerable distance (∼50 Å) from the CD. Instead of interacting with the CD of the trimer they belong to, for the majority of the P8 molecules the PDs swap between the trimers, each PD interacting with the CDs and PDs of adjacent trimers via heterotypic interactions

(Fig. 2e,f). Nine such structures are observed in the icosahedral asymmetric unit of the NC.

In addition to the 'open' conformation described above and adopted by 9 out of 10 of the P8 chains, a fundamentally different conformation is also found on the shell. This 'closed', more compact, conformation is observed in one of the three chains of the peripentonal $P8_Q$ trimer (Fig. 2g; Supplementary Fig. 2). The two conformations differ in degree of bending of the elongated linker that acts as a hinge. The change in relative position between the two conformations is ∼75 Å and involves a rotation of 71 degrees (Fig. 2h). In the closed conformation, the PD is bound to its own CD via a homotypic interaction (Fig. 2i). This interaction is characterised by an interface that is analogous to the one created between the PD and a CD of an adjacent trimer in heterotypic interactions (Fig. 2j).

**P4 hexamer is anchored to the vertices of the P1 shell**. In the members of the subfamily *Sedoreovirinae* of the family *Reoviridae*, such as blue tongue virus (*Orbivirus*)[29] and rice dwarf virus (*Phytoreovirus*)[32], P-type trimers occupy the five-fold vertices. In contrast, in members of *Cystoviridae*, such as φ6 (ref. 23) and *Pinareovirinae*, such as aquareoviruses[27], the P-type trimers are absent leaving room for turret-like structures at the icosahedral five-fold vertices. In φ6, the vertices are occupied by the P4 packaging hexamer (Fig. 1). The presence of P4 hexamers at the five-folds provides also a rationale for the observed closed conformation of certain peripentonal P8 proteins (Supplementary Fig. 3); if the type $P8_Q$ monomers were all in the open conformation, significant clashes would occur between $P8_Q$ and P4 (Supplementary Fig. 3b).

The six-fold symmetry of the P4 hexamer and five-fold symmetry of the icosahedral vertex beneath creates a symmetry mismatch, so that the structure of P4 cannot be resolved using conventional icosahedral reconstruction[23]. To address this we applied the localized reconstruction method[17] to P4 hexamers. Local areas corresponding to P4 hexamers were extracted from the NC images and reconstructed as single particles with six-fold symmetry ('P4 hexamer reconstruction') or without any symmetry ('NC vertex reconstruction'). This allowed a low-resolution structure of the P4 hexamer to be determined *in situ* (Fig. 3). At a resolution of 9.1 Å (FSC = 0.143), fitting of a P4 crystal structure (PDB:4BLO)[19] into the density was unambiguous (Fig. 3a–c) and allowed us to orient the hexamer relative to the NC vertex (Fig. 3d,e).

A disorder prediction algorithm suggests that the C-terminal residues of P4 are likely to be disordered (Supplementary Fig. 4)[35]. Indeed, they are missing in the crystal structure and this disordered region has been proposed to attach the P4 to the underlying P1 shell[19]. However, in earlier low-resolution cryo-EM reconstructions, the P4 hexamers appeared to float on top of the five-fold vertexes of the P1 shell[23,36,37] and the connections between these symmetry-mismatched components have remained unresolved. Our localized reconstruction of P4 hexamers that correctly dealt with the symmetry mismatch revealed elongated densities extending from 3 of the 6 P4 C-terminal parts (Fig. 3f). We hypothesised that these may correspond to the C-terminal parts that are disordered in the crystal structure. Consistent with this hypothesis, in the high-resolution icosahedral NC reconstruction we observed sixty copies of elongated density that lacked any apparent secondary structure stretching ∼115 Å on top of each of the asymmetric P1 dimers (Figs 1b and 4c–e). At the vertex distal part of this density, densities consistent with the side-chains 324-ProArgArg-326

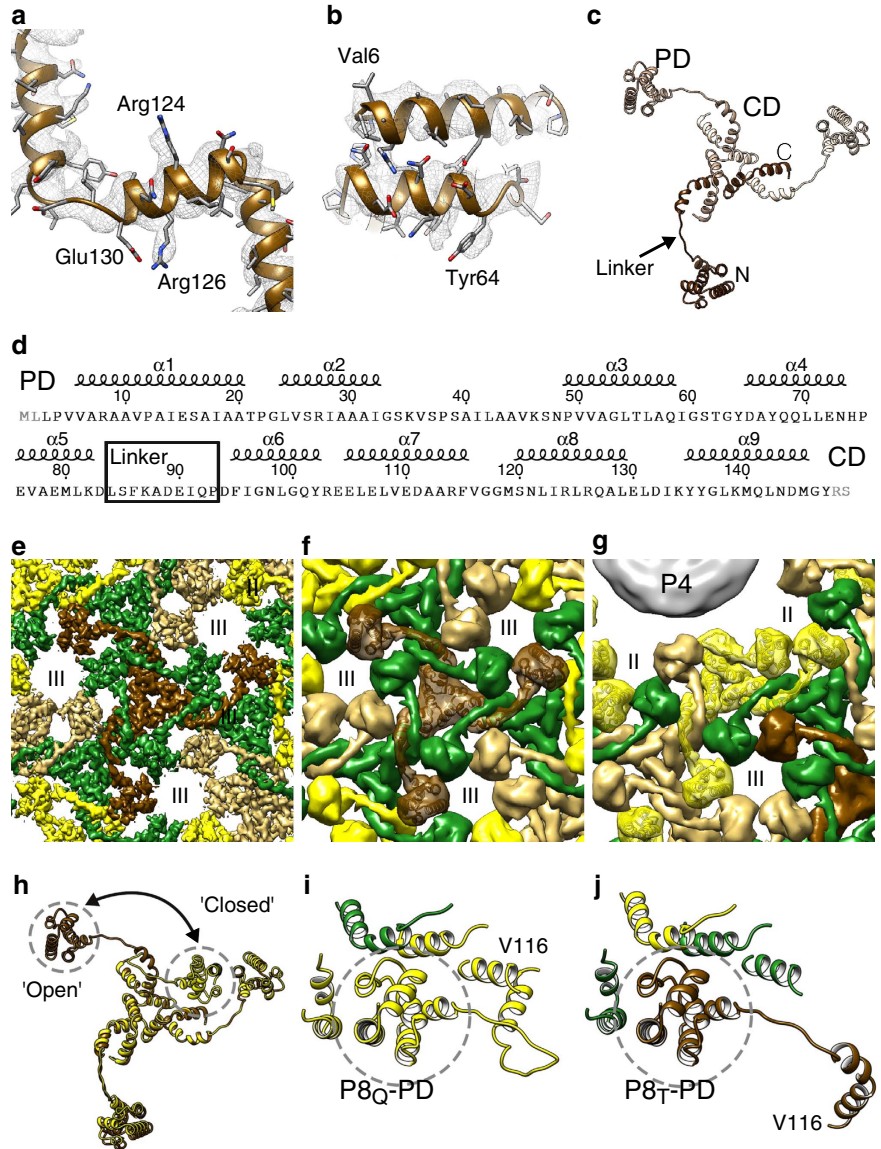

**Figure 2 | Domain swapped architecture of the outer φ6 P8 shell.** (**a,b**) Close-ups of the atomic model of the T-type trimer ($P8_T$) and corresponding density (mesh). Some side chains are labelled. (**c**) Atomic model of the P8 trimer. The three different chains are shown in different shades. The peripheral domain (PD) and the core domain (CD) are labelled and the corresponding areas of the structure circled. The linker (arrow), N terminus (N) and C terminus (C) are labelled for one chain (dark brown). (**d**) Sequence of the P8 with secondary structure elements, PD, CD and the linker region labelled. Amino-acid residues not resolved in the structure are in grey. (**e**) A close-up of the P8 shell density viewed along the three-fold axis of symmetry. The linkers from different trimers intertwine, leading to domain swapping of the peripheral domains between the trimers. (**f**) Same view as in **e** but a rough molecular surface is shown for clarity. $P8_T$ is shown as a ribbon. (**g**) Same as **f** but a view showing the Q-type trimers ($P8_Q$, ribbon) around the P4 hexamers. P4 is shown as a grey surface. (**h**) Structural alignment of $P8_T$ and $P8_Q$ trimers. The location of the peripheral domain in 'open' and 'closed' conformation of the trimer is indicated. (**i**) The mode of interaction of $P8_Q$-PD in closed conformation is shown. The interactions are mainly of homotypic nature with other parts of the same trimer. (**j**) The mode of interaction of the $P8_T$-PD in open conformation is shown. The interactions are of heterotypic nature with other types of trimers. The P8 trimers are coloured in yellow ($P8_Q$), green ($P8_R$), gold ($P8_S$) and brown ($P8_T$) and the type II and type III holes[23] are labelled. See also Supplementary Figs 2 and 3.

of the P4 C terminus were identified (Fig. 4f), allowing us to specifically assign this density to the C-terminal residues 292–332 of P4. The sixth, unbound C-terminal region of P4 was not resolved in any of the reconstructions, and is likely to be either disordered or unresolved in our maps due to limited resolution of the localized reconstructions.

Binding of up to five of the six flexible C-terminal tails of P4 provides an elegant solution to the symmetry mismatch. These tails are likely to be disordered in solution (Supplementary Fig. 4) and to order only upon binding to P1 during assembly.

The P4 tails are analogous to the 'stay-cables' observed earlier in adenovirus fibres[38]. As such cables have now been observed in at least two unrelated viruses, adenovirus (a dsDNA virus)[38] and φ6 (a dsRNA virus, this study), they may be a general solution to the problem of attaching fibres and other symmetry-mismatched appendices to vertices of icosahedral virus capsids.

**P4 C-terminal tail solidifies the dimeric P1 building block.** The inner P1 shell is composed of 60 copies of asymmetric P1 dimers,

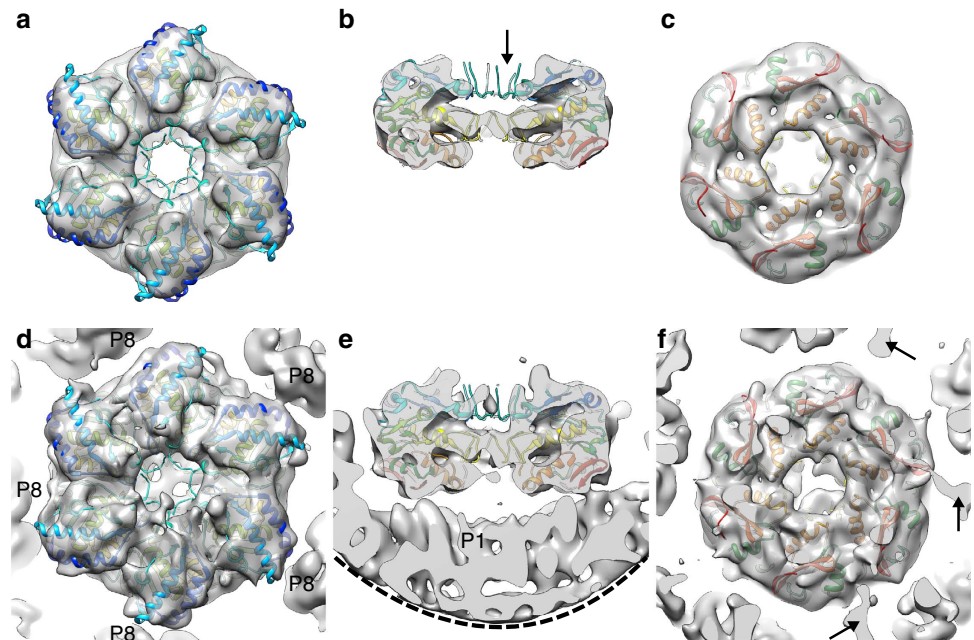

**Figure 3 | Hexameric packaging ATPase P4.** (**a–c**) Localized reconstruction of ϕ6 P4 with six-fold symmetry is shown as a grey transparent surface from the top (**a**), side (**b**) and below (**c**). P4 crystallographic structure[19] is coloured from red (C-terminus) to blue (N-terminus) and was fitted into the reconstruction as a rigid body. Flexible loops that were only partially resolved in the crystallographic structure were not resolved in the localized reconstruction (arrow). (**d–f**) Same views as in **a–c** showing the localized reconstruction without symmetry. The edge of a mask cutting through the density under the P4 hexamer is indicated with a dashed line in **e**. Three densities possibly connecting the P1 shell to the P4 C-termini are indicated with arrows.

formed by P1$_A$ and P1$_B$ (see Fig. 1b)[23]. A large-scale expansion of the P1 shell occurs during RNA packaging to empty particles and subsequent RNA replication[22,23]. To model the structural transitions in the P1 shell during this expansion, we refined the existing atomic coordinates of the P1 monomer (PDB:4K7H)[39] into our EM density map of the NC (Fig. 4a,b) and compared them to P1 coordinates flexibly fitted to an unexpanded PC reconstruction (PDB:4BTG)[39]. The angle between the two subunits in the P1$_A$–P1$_B$ asymmetric dimer changes relatively little (11°) showing that the P1$_A$–P1$_B$ dimer behaves as quite a rigid block during the expansion (Supplementary Fig. 5a,b; Supplementary Movie 1). In contrast to these intradimer angles, the interdimer angles change significantly, mainly at the P1$_B$–P1$_B$ interfaces[39]. The interdimer angle changes 57° at the two-fold interface and 49° at the three-fold interface (Supplementary Fig. 5c,d). The root-mean-square deviation (RMSD) calculated between the two states was 2.6 and 6.0 Å for the P1$_A$ and P1$_B$ C-alpha coordinates, respectively. This shows that relatively small changes occur within the P1$_A$ subunits, while P1$_B$ subunits adjust to their local environment in the two very different P1 shell expansion states.

The relative rigidity of the P1$_A$–P1$_B$ dimer interface is now explained by our observation that the C-terminal part of P4 bridges the P1$_A$ and P1$_B$ monomers (Fig. 4c,d) whilst still allowing interdimer angles to change. Thus, the P4 C-terminus seems not only to link P4 hexamers to the P1 shell, but also to play a role in P1 dimerization. A dimer of the major inner capsid protein could indeed be a conserved intermediate in the assembly pathways of dsRNA viruses. This notion is supported by the structure of the *Penicillium chrysogenum* virus in which the equivalent building block is formed by a covalently linked major capsid protein dimer, produced by gene duplication[40]. In ϕ6, we propose that the C-terminal tail of P4 provides an analogous linkage.

## Discussion

When combined with earlier structural and biochemical results, our NC structure allows us to propose an assembly model for ϕ6 (Fig. 5; Supplementary Movie 2). The assembly of the P1 shell is initiated *in vitro* by P4 hexamers[26] and the C-terminus of P4 is essential for the formation of recombinant PC *in vivo*[41]. We now have a structural rationale for these observations, since the P4 C-terminus cements the P1 asymmetric dimer together (Fig. 4c,d). Accordingly, we suggest that the ϕ6 PC assembly starts with the formation of a P1$_A$–P1$_B$ dimer stabilized by the C-terminus of P4 (Fig. 5a). As observed in the P1 crystal structure, individual P1 chains tend to create a pentamer in a conformation that resembles a group of five P1$_A$ chains in the unexpanded conformation of PC[39]. Our modelling experiments also showed that six P1$_A$–P1$_B$ dimers per P4 would create steric clashes (Supplementary Movie 2). It is thus likely that each P4 grabs five and not more P1$_A$–P1$_B$ dimers. Dimers attached to distinct P4 hexamers then interact to eventually form a closed P1-shell (Fig. 5b). The assembly of the compact empty PC is facilitated by the assembly cofactor P7 (refs 20,26). P7, together with P2, is located at the three-fold symmetry axis of the empty PC[21], but the order of P2 and P7 binding to the PC remains unclear. The PC subsequently packages and replicates the RNA genome segments leading to expansion of the PC (Fig. 5c)[23,39] and finally formation of the double-shelled particles by incorporation of the P8 layer (Fig. 5d).

P8 shells form spontaneously in the presence of 0.1–1.0 mM Ca$^{2+}$ and in the absence of any other structural proteins[42]. Conversely, depleting calcium by a chelating agent or exposure to low pH leads to P8 shell disassembly[42,43]. Raman spectroscopy has revealed that the assembled shell stabilizes the mostly α-helical conformation of P8 (ref. 44). It is thus clear that Ca$^{2+}$ regulates the conformational state of P8. In the light of our observation of a closed and open state of the P8 trimer in

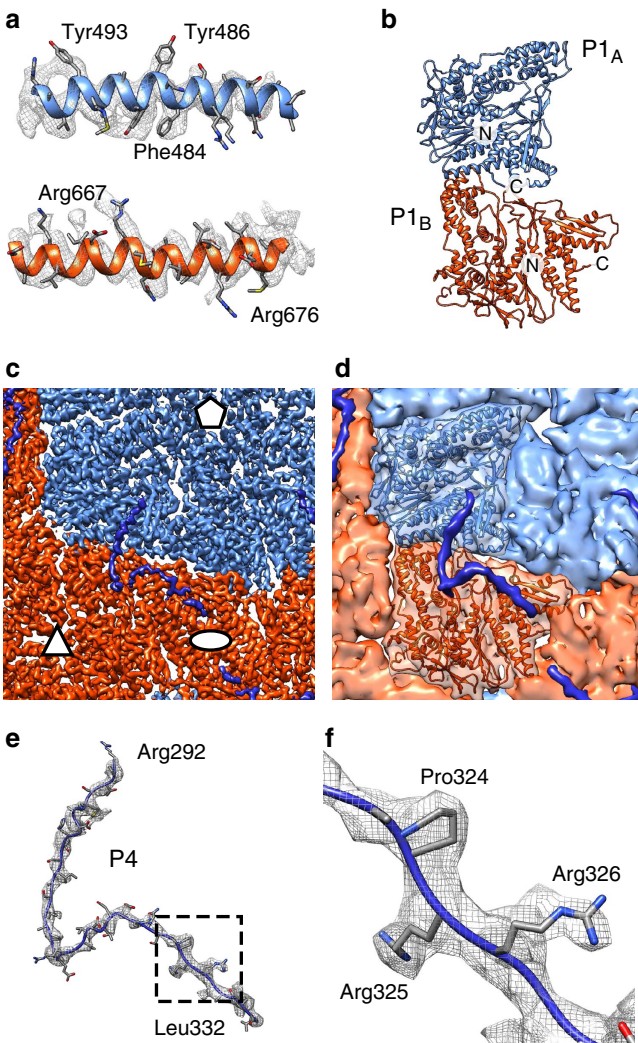

**Figure 4 | Structure of the inner P1 shell.** (**a**) Close-ups of the atomic models for the φ6 P1$_A$ (top) and P1$_B$ (bottom) subunits and corresponding density (mesh). (**b**) Atomic model of the asymmetric P1 dimer. The two subunits and the N- and C-termini are labelled. (**c**) Close-up of the density map. Icosahedral symmetry axes are labelled with an ellipse (two-fold), a triangle (three-fold) and a pentagon (five-fold). (**d**) Same view as in **c** but a rough molecular surface is shown for clarity and one P1 dimer is shown as a ribbon. P1$_A$ is blue, P1$_B$ is red, and density assigned to P4 C-terminal part is dark blue. (**e**) Atomic model for the P4 C-terminal part (R292–L332) and the corresponding density (mesh) are shown. (**f**) Close-up of the area indicated in **e** showing the 324-ProArgArg-326 motif that allowed assignment of this density to P4. See also Supplementary Fig. 4 and Supplementary Movie 1.

the NC, we hypothesize that Ca$^{2+}$ induces transition of the P8 trimer from the closed to the open conformation and this further facilitates outer shell assembly by domain swapping (Fig. 5d,e). Due to steric clashes, the P8 monomers closest to the P4 hexamers are forced to remain in closed conformation (Supplementary Fig. 3). Ca$^{2+}$ regulation of P8 conformation, possibly in addition to acidic pH, might also be important to facilitate the delivery of the PC to the host cell upon infection.

The conformational space sampled by P8 trimers in solution and at different Ca$^{2+}$ concentrations remains unknown. Insolubility of purified P8 and its tendency to readily form large assemblies even in the presence of Ca$^{2+}$-chelating agents has impeded its solution structure studies by small angle X-ray

scattering (Z.S. and J.T.H., unpublished observations). The P8 CD domain is preceded by Pro93 that breaks the helix α6 of the CD. Proline residues are abundant in the linker regions of domain swapped proteins and may play a role in modulating the equilibrium between an open and a closed state[45]. It is thus possible that Pro93 has a functional role in adding strain to the backbone of the linker region but this hypothesis remains to be tested.

The observed P8 shell assembly allows us to extend the concept of domain swapping to viral shells[8–12]. Domain swapping is considered *bona fide*, when the same subunit can still adopt both closed and open conformations, as opposed to the subunits adopting just the open conformation in a multimer and the closed conformation in a different, albeit orthologous, monomeric protein[8]. Following this nomenclature, our φ6 outer shell structure provides an exemplar of *bona fide* domain swapping between P8 trimers that exhibit both closed and open conformations.

Analogously to monomer to multimer transition during the evolution of multimeric proteins, our results allow us to propose a hypothetical model how closed viral protein shells may further evolve from such proteins. A primordial, relatively simple capsomer may have some tendency to form larger assemblies, but initially the protein-protein interfaces required to make a spherical shell are not optimized. Mutations in a linker region can increase its flexibility to allow for the formation of an open conformation of the multimeric capsomer. These capsomers in their open conformations would then be able to further link together by recapitulating the domain-domain interactions of the closed conformation. A pre-existing inner protein shell, genome, or internal lipid bilayer present in some viruses, may play a role in increasing the fidelity of shell assembly[46]. This is also evident in the case of φ6, where in the absence of the inner P1 shell, P8 self-assembles *in vitro* into aberrant, shell-like structures that vary in size and are not topologically closed[43,47]. In some viruses, the capsomers may have subsequently lost the ability to form both open and closed conformations. To what extent the outer protein shells of φ6 and other dsRNA viruses are homologous remains an open question, as domain-swapping is not observed in other dsRNA viruses. It is conceivable, however, that linker region deletions and domain-shuffling may have played a further role in stabilizing protein shells. In the case of φ6, the ability of P8 to still adopt both conformations may have been preserved in evolution as this may play a role in calcium-regulated outer shell assembly/disassembly and also the presence of P4 hexamer at the vertices necessitates a closed conformation of P8.

In conclusion, our structural characterization of the φ6 double-shelled particle revealed interactions of the inner P1 shell and symmetry-mismatched P4 packaging hexamers. Most importantly, the study led to the discovery of two states of the outer shell forming protein P8, 'open' and 'closed'. Trimers in the 'open' conformation swap domains between each other. This process is possibly regulated by calcium ion concentration but further studies are required to reveal the exact mechanism of Ca$^{2+}$ regulation. The derived outer shell assembly model not only provides insights into assembly in this particular model system, but also suggests how closed protein shells may have arisen during evolution. Further studies are required to assess to what extent domain swapping may have played a role in other viral systems.

## Methods

**Cryo-EM sample preparation.** Bacteriophage φ6 was grown on its host *Pseudomonas syringae pv.phaseolicola* HB10Y[48] and purified as described previously[49]. The viral envelope was removed by Triton X-114 extraction[49] and the resulting aqueous mixture containing the NC loaded onto a CIM DEAE-1

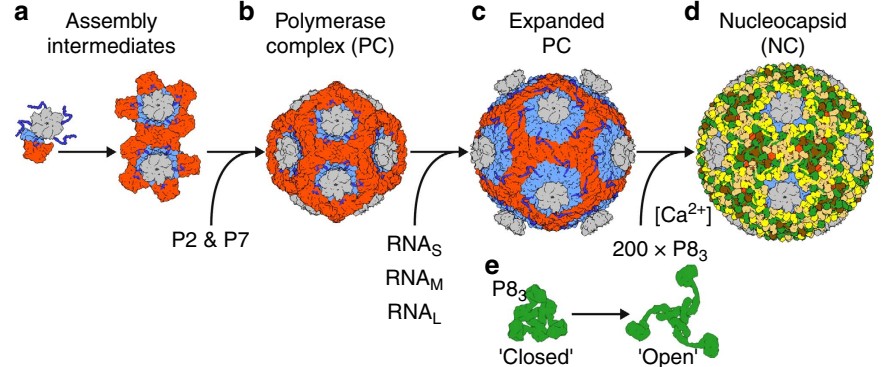

**Figure 5 | Assembly model. (a)** The ϕ6 inner capsid is composed of protein P1, P2, P4 and P7. Preassembled P4 hexamers (grey) nucleate the assembly of the P1 shell by bringing together two P1 subunits that will form the asymmetric P1$_A$–P1$_B$ dimer (light blue and red) in the final assembly. The dimerization of P1 is facilitated by the C-terminal tail of P4 (dark blue). Up to five P1 dimers can assemble around a P4 hexamer. **(b)** P1 dimers connected to distinct P4 hexamers interact bringing together the precursors of the icosahedral vertices. This is facilitated by protein P7, which acts as an assembly cofactor. **(c)** The self-assembled empty capsids package and replicate the single-stranded RNA precursors of the ϕ6 dsRNA genome (S, M and L) and the capsid adopts the expanded conformation. **(d)** Trimers of P8 (green, yellow, gold and brown) then assemble around the dsRNA containing inner capsid to form the nucleocapsid. **(e)** The assembly of the P8 shell is induced by Ca$^{2+}$ ions and involves major conformational changes in the linker regions of P8 subunits to mediate the transition from the closed to the open conformation. See also Supplementary Movie 2.

**Table 2 | Data processing parameters.**

| Parameter | NC particle | P4 hexamer | NC vertex |
|---|---|---|---|
| Particles* | 13,291 (16,466) | 56,448 (159,492) | 56,448 (159,492) |
| Box size (pixels) | 512 | 128 | 128 |
| Pixel size (Å) | 1.35 | 1.35 | 1.35 |
| Symmetry | I1 | C6 | C1 |
| Resolution (Å) | 4.0 | 9.1 | 9.1 |
| B-factor (Å$^2$)† | − 177 | − 600‡ | − 600‡ |

NC, nucleocapsid.
*Number of particles (or subparticles in the case of P4 hexamer and NC vertex) chosen for averaging the final map is given. The number of all particles extracted from the micrographs (or all subparticles extracted from particles) is given in parenthesis.
†Inverse B-factor used for sharpening of the map.
‡Value of B-factor determined *ad hoc*.

tube monolithic column (BIA Separations, Slovenia) in 20 mM potassium phosphate pH 7.2, 1 mM MgCl$_2$, 0.1 mM CaCl$_2$, 150 mM NaCl from which the bound NC particles were eluted using a 0.15–2 M NaCl complex gradient (in 20 mM potassium phosphate pH 7.2, 1 mM MgCl$_2$, 0.1 mM CaCl$_2$). The NC containing fractions were concentrated using an ultra-centrifugal filter with a 100-kDa cutoff (Amicon; EMD Millipore, Billerica, MA, USA). A 3-µl aliquot of sample, diluted in buffer (20 mM KPO$_4$, pH 7.2, 1 mM MgCl$_2$, 0.1 mM CaCl$_2$, 150 mM NaCl) to concentration of 3 mg ml$^{-1}$, was applied to a glow-discharged grid (C-flat; Protochips, Raleigh, NC) and vitrified by plunge-freezing into liquid ethane using a Vitrobot (FEI, Hilsboro, OR).

**Data acquisition and processing.** Data were acquired using a 300-kV transmission electron microscope (Tecnai F30 'Polara'; FEI) equipped with an energy filter (slit width 20 eV; GIF Quantum LS, Gatan, Pleasanton, CA) and a direct electron detector (K2 Summit, Gatan). Movies (22 frames, each frame 0.2 s) were collected at in electron counting mode at dose rate of 5 e$^-$ per pixel per s at calibrated magnification of 37,037 × resulting in a total dose of ∼16 e$^-$ per Å$^2$ and pixel size of 1.35 Å. Data acquisition statistics are summarized in Table 1.

Movie frames were aligned to account for drift[50] and contrast transfer function (CTF) parameters were estimated[51]. A total of ∼900 movies, showing minimal drift and astigmatism were used to pick 16,466 particles. Particles were extracted for 2D reference free classification in Relion 1.3 (ref. 52). Particles from classes showing complete P1 and P8 layers were selected for 3D classification using a previously published NC structure (EMD-1,206) as an initial model. A subset of 13,291 particles extracted from movies was refined using standard refinement and particle polishing protocols implemented in Relion.

The structure of the P4 hexamer was calculated using localized reconstruction (www.opic.ox.ac.uk/localrec)[17]. Briefly, twelve 'subparticles', each corresponding to a projection of the P4 hexamer in each of the particle images, were extracted (total of 159,463 subparticles) to a smaller box and treated as independent single particles. To assign the orientation for each subparticle, one of the five possible

triplets of Euler angles that were equivalent for the corresponding five-fold symmetric vertex was chosen randomly so as not to bias orientations in further processing. Two types of reconstructions were calculated. For the first reconstruction ('P4 hexamer'), partial signal subtraction was used to remove the contribution of unwanted NC density in the particle images[53]. We modified our earlier localized reconstruction method[17] to allow subtracting 'all-but-one' P4 hexamer density (available from http://github.com/OPIC-Oxford/localrec), in addition to the other unwanted densities (P1 and P8 shells and RNA) from the icosahedral reconstruction. Only subparticles from the edge of the particle (total of 103,482; maximum 40° deviation from a side view) were chosen for further analysis to reduce the overlap of the projected hexamer density with the RNA component of the virion that could not be accurately subtracted from the particle images. The structure of the P4 hexamer was then determined using Relion and 3D classification, using a localized reconstruction of the hexamer, calculated from a subset of subparticles and filtered to 40 Å resolution, as a starting model. 3D classification revealed the hexamer in different orientations relative to the five-fold vertex. Particles from classes showing clear hexamer density were combined and six-fold symmetry was applied on the reconstruction. For the second reconstruction ('NC vertex'), subparticles extracted from the original particle images and orientation parameters from the P4 hexamer reconstruction were used and no symmetry was applied.

Resolutions of the icosahedral NC reconstruction and P4 hexamer localized reconstruction were estimated by using FSC using 0.143 cutoff. The NC vertex reconstruction was low-pass filtered to the resolution of P4 hexamer reconstruction (9.1 Å) and both localized reconstructions were sharpened by applying an inverse B-factor of − 600 Å$^2$ (determined by trial and error). Local resolution of the NC reconstruction was estimated using ResMap[54]. Reconstruction statistics are summarized in Table 2.

**Atomic model building and refinement.** As no P8 atomic model was available, a polyalanine model was built *de novo* using Coot. Visually identified densities

**Table 3 | Atomic model refinement statistics.**

| Parameter | P1 | P8 |
|---|---|---|
| *Refinement* | | |
| Resolution (Å) | 4.0 | |
| Map CC | 0.715 | |
| *RMS deviations* | | |
| Bonds (Å) | 0.09 | |
| Angles (°) | 1.0 | |
| *Molprobity validation* | | |
| Clashscore, all atoms (percentile) | 13.8 (55th) | 10.4 (70th) |
| Molprobity score | 1.92 (80th) | 1.53 (94th) |
| Favoured rotamers (poor) (%) | 98.4 (0.1) | 97.3 (0.0) |
| Ramachandran (% favoured) | 96 | 99.0 |
| Ramachandran (% allowed) | 99.7 | 100 |
| Ramachandran Plot (% outliers) | 0.3 | 0 |

RMS, root mean square.

for bulky residues facilitated building the full atomic model of P8 (L3-Y147) and the C terminus of P4 (R292-L332) with side-chains. The crystal structure of φ6 P1 (PDB:4K7H) was fitted in the NC map using COOT[55] as a rigid body in two different positions corresponding to subunits P1$_A$ and P1$_B$. P1$_A$ and P1$_B$ main-chains and side-chains were adjusted using manual and real space fitting in COOT. The crystal structure of φ6 P4 (PDB:4BLO)[19] was fitted in the P4 hexamer and NC vertex localized reconstructions as a rigid body in UCSF Chimera[56].

Atomic models for all of the chains in the asymmetric unit (P1$_A$, P1$_B$, P4 C terminus and ten copies of P8) were refined in Phenix.real_space_refine applying secondary structure, rotamer, and Ramachandran plot restraints. In addition, non-crystallographic symmetry restraints were applied on P8 chains[57]. The models were validated with MOLPROBITY[58]. Atomic model refinement statistics are summarized in Table 3. Disordered regions in P4 sequence (UniProt:P11125) were predicted with RONN[35] and changes in subunit-subunit angles calculated in UCSF Chimera[59].

**Data availability.** Density maps and atomic models that support the findings of this study have been deposited in the Electron Microscopy Database and in the Protein Databank with the accession codes EMD-3571 (NC reconstruction), PDB 5MUU (NC atomic model), EMD-3572 (P4 hexamer reconstruction), PDB 5MUV (P4 atomic model fitted in the P4 hexamer reconstruction), EMD-3573 (NC vertex reconstruction), and PDB 5MUW (P4 atomic model fitted in the NC vertex reconstruction).

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

## Acknowledgements

We thank Alistair Siebert for electron microscopy support and Riitta Tarkiainen for technical help. The OPIC electron microscopy facility was founded by a Wellcome Trust JIF award (060208/Z/00/Z) and is supported by a WT equipment grant (093305/Z/10/Z). The EU ESFRI Instruct Centre for Virus Production (ICVIR) facility used in this study is supported by the Academy of Finland (grant 272853) and University of Helsinki and this is a collaboration with the Oxford Instruct Centre. This work was funded by Academy of Finland (218080 and 263677 to J.T.H; 283192, 250113, and 272507 to M.M.P.), Sigrid Jusélius Foundation, Medical Research Council (MR/N00065X/1 to K.E.O.), European Research Council under the European Union's Horizon 2020 research and innovation programme (649053 to J.T.H.) and Wellcome Trust Core Award Grant Number 090532/Z/09/Z. The authors acknowledge the support of employees and the use of experimental resources of Instruct, a Landmark ESFRI project.

## Author contributions

Z.S., S.L.I. and J.T.H. collected and analysed cryo-EM data. K.E.O., M.M.P., D.I.S. and J.T.H. designed the study. X.S. prepared the samples. Z.S., K.E.O. and A.K. built and refined the atomic models. Z.S. and J.T.H. wrote the manuscript and all authors read and commented on the manuscript.

## Additional information

**Competing interests:** The authors declare no competing financial interests.

