## [Peer Review File · Nature Communications]

Reviewers' Comments:

Reviewer #1 (Remarks to the Author):

The authors present relatively high resolution cryoEM data of the bacteriophage $\phi 6$. They show that the P8 protein, forming the out shell of the virus, exists in domain swapped trimers. These findings are novel and will be of high interest to researchers across disciplines including virology and structural biology. How an individual protein can adopt two different conformations, yet still pack within the confines and restraints of a large icosahedral virus is remarkable.

General questions/points requiring clarification:

1. Could the authors show density for the linker region between the two domains (PD and CD) of the P8 protein in both the open and closed conformations? This could be incorporated into Figure 2.
2. This is important to unequivocally demonstrate that these domains belong to the same monomeric protein.
2. There were 3 class averages identified from the cryoEM data. Some of these classes were "empty" in terms of RNA packaging. Was the resolution of these other classes sufficient to test whether these domain swapped dimers were also present in empty VLPs?
3. The authors note that proline residues can modulate domain swapping conformations, and identify Pro93 preceding the CD domain. Do the open and closed conformations of P8 differ substantially around this area? Is there a cis-trans change for example? Related to point 1, can density be shown for this area?
4. How conserved is the Pro93 in these viruses? Whilst it would be interesting to test the importance of this residue (and indeed the length of linker region) through mutation, I don't believe this is necessary to support the conclusions of the paper.
5. Line 99: The P8 subunits are mostly in an open conformation, swapping domains with their neighbours. Can "mostly" be swapped with a value indicating the proportion of P8 trimers which exist in open and closed conformations?

Jade Forwood

Reviewer #2 (Remarks to the Author):

This is a rather easy referee report to write, as the manuscript by Sun et al is excellent and I recommend publication in Nature Communications. I do not think there are any substantive questions to be address by the authors: the manuscript and figures are excellent, and clearly describe two interesting and significant results: domain swapping and the symmetry mismatch between the capsid and P4, which is nicely done.

I have two comments that are intended to try and increase the accessibility of the manuscript to a non-specialist reader.

The first is that in the introduction, I think there's a slight disconnect between lines 69 and 70. An interesting story about domain swapping is developed and then appears to be set aside as they move to phi6. The link between the two is a little bit opaque, especially to the non-specialist reader. Is there a reason to think this is a good model system to study domain swapping? If so it should be expanded upon or the order of text amended to make a cleaner logical flow of information.

The second suggestion is re figures, which are generally excellent, particularly in that the colour schemes are clear and consistent between figures, easing the development of ideas across the figures. These must be made as large as possible in the final formatted paper (for Nature Comms). I would also suggest that the authors look at their use of silhouette in chimera (presumably). This makes several of the figures very dark and the colour scheme muddy - eg fig 1a/b and fig2 (especially 2h). Removing or weakening this effect might lift the figure and make it easier to see what is going on.

Minor typos:

I only found two instances where an additional "the" might be inserted to aid clarity:

Line 70: "...the Cystoviridae family"?

Line 203, "...of the P4 C-terminus"?

Reviewer #3 (Remarks to the Author):

This is an interesting paper describing the striking high-resolution structure of the phi6 nucleocapsid, a T=13 shell resembling other dsRNA viruses. The structures have been determined to 4Å resolution using cryo-EM and the maps are of excellent quality, allowing most of the icosahedrally organized proteins to be modeled with high accuracy.

One of the most striking observations is that the P8 trimer has two domains that can switch into dramatically different conformations via a central hinge region. The authors call it “domain swapping” but I’m not sure this is the right term, since in traditional domain swapping, a domain from one subunit would replace a similar domain from another subunit. In this case, the “domain swapped” PD takes up a unique position not otherwise occupied by a similar domain from another subunit. I would rather refer to it by the more general term “conformational switching.” This terminological quibble aside, it is a fascinating phenomenon that also provides some insights into the possible assembly pathway of this shell (and by inference, T=13 shells in other viruses).

The authors’ musings on the evolutionary implications of their structure are interesting. It is well known, of course, that conformational switching is important to build large icosahedral shells, but I think they are overselling the generality of the process. As they point out, even other T=13 dsRNA viruses do not appear to have a similar switch.

Otherwise, the paper is well written and figures look nice. However, it is hard to figure out from Fig 2 which subunits correspond to Q, R, S and T and exactly how the trimers are organized in the context of the icosahedral asymmetric unit. A schematic diagram would be helpful.

Line 132: leavo → laevo.

REVIEWERS' COMMENTS:

Reviewer #1 (Remarks to the Author):

The authors present relatively high resolution cryoEM data of the bacteriophage $\phi 6$. They show that the P8 protein, forming the out shell of the virus, exists in domain swapped trimers. These findings are novel and will be of high interest to researchers across disciplines including virology and structural biology. How an individual protein can adopt two different conformations, yet still pack within the confines and restraints of a large icosahedral virus is remarkable.

General questions/points requiring clarification:

1. Could the authors show density for the linker region between the two domains (PD and CD) of the P8 protein in both the open and closed conformations? This could be incorporated into Figure 2. This is important to unequivocally demonstrate that these domains belong to the same monomeric protein.

We are thankful for this opportunity to add this important detail. We have now added close-ups of three representative linker regions (one in a closed conformation and two in an open conformation) as a new Supplementary Figure 2. We have also labelled Pro93 (as requested in point 3). We would like to note that the linker region density is very clear in both conformations and the path of the polypeptide chain is unambiguous.

2. There were 3 class averages identified from the cryoEM data. Some of these classes were "empty" in terms of RNA packaging. Was the resolution of these other classes sufficient to test whether these domain swapped dimers were also present in empty VLPs?

The particles assigned to the 'empty' class correspond most likely to broken particles that have leaked out the dsRNA during or after purification (particle expansion and P8 shell assembly occur only after RNA packaging). They are thus most likely not biologically relevant. Nevertheless, as suggested, we have now reconstructed a map using all of the 589 particles assigned in this 'empty' class. The resolution of the reconstruction was limited to 6.7 Å, presumably due to the relatively small number of particles and structural heterogeneity. This resolution was not sufficient to address reviewer's question.

3. The authors note that proline residues can modulate domain swapping conformations, and identify Pro93 preceding the CD domain. Do the open and closed conformations of P8 differ substantially around this area? Is there a cis-trans change for example? Related to point 1, can density be shown for this area?

As suggested, we are now showing the density for this area in a new Supplementary Figure 2 (please see our answer to point 1). The peptide bond preceding Pro93 residue is consistent with a trans configuration in both open and closed conformation. We have thus no evidence of cis-trans change taking place.

4. How conserved is the Pro93 in these viruses? Whilst it would be interesting to test the importance of this residue (and indeed the length of linker region) through mutation, I don't believe this is necessary to support the conclusions of the paper.

We have now aligned the Phi-virus P8 sequences from phi6, phi7, phi13 and phiNN (the only other available P8 sequences, i.e. those of phi8, phi12 and phi2954, were too dissimilar in their sequence and length to be added to the alignment). All four have a proline in position 93. However, we would like to note that three of these (phi6, phi7 and phiNN) are very similar in their sequence (89–97% identity) so the alignment is not very informative. The fourth (phi13) is quite dissimilar (25% identity) so the alignment of phi13 P8 relative to the other three may not be very accurate. Thus we would prefer not to comment on the conservation of Pro93, as this might be misleading. More complete sequences would be needed for this analysis, but this is outside of the scope of our current paper.

5. Line 99: The P8 subunits are mostly in an open conformation, swapping domains with their neighbours. Can "mostly" be swapped with a value indicating the proportion of P8 trimers which exist in open and closed conformations?

The text has been modified as follows: “The majority of the P8 subunits (540 of 600) are in an open conformation, swapping domains with their neighbours”

Jade Forwood

Reviewer #2 (Remarks to the Author):

This is a rather easy referee report to write, as the manuscript by Sun et al is excellent and I recommend publication in Nature Communications. I do not think there are any substantive questions to be address by the authors: the manuscript and figures are excellent, and clearly describe two interesting and significant results: domain swapping and the symmetry mismatch between the capsid and P4, which is nicely done.

I have two comments that are intended to try and increase the accessibility of the manuscript to a non-specialist reader.

The first is that in the introduction, I think there's a slight disconnect between lines 69 and 70. An interesting story about domain swapping is developed and then appears to be set aside as they move to phi6. The link between the two is a little bit opaque, especially to the non-specialist reader. Is there a reason to think this is a good model system to study domain swapping? If so it should be expanded upon or the order of text amended to make a cleaner logical flow of information.

Before starting the study, we had no reason to believe that phi6 would be a good model system to study domain-swapping.

The second suggestion is re figures, which are generally excellent, particularly in that the colour schemes are clear and consistent between figures, easing the development of ideas across the figures. These must be made as large as possible in the final formatted paper (for Nature Comms). I would also suggest that the authors look at their use of silhouette in chimera (presumably). This makes several of the figures very dark and the colour scheme muddy - eg fig 1a/b and fig2 (especially 2h). Removing or weakening this effect might lift the figure and make it easier to see what is going on.

We are very grateful for the recommendation to provide as much space as possible for the figures. As the structure is very complicated due to domain swapping, we added the silhouettes for extra clarity because without them, the figures were confusing. In the high-resolution versions of the figures the silhouettes do not seem to cause any problem and thus we would not expect this to happen in the final formatted paper either. However we are happy to modify the figures as needed and advised by the production staff.

Minor typos:

I only found two instances where an additional "the" might be inserted to aid clarity:

Line 70: "...the Cystoviridae family"?

Corrected.

Line 203, "...of the P4 C-terminus"?

Corrected.

Reviewer #3 (Remarks to the Author):

This is an interesting paper describing the striking high-resolution structure of the phi6 nucleocapsid, a T=13 shell resembling other dsRNA viruses. The structures have been determined to 4Å resolution using cryo-EM and the maps are of excellent quality, allowing most of the icosahedrally organized proteins to be modeled with high accuracy.

One of the most striking observations is that the P8 trimer has two domains that can switch into dramatically different conformations via a central hinge region. The authors call it "domain swapping" but I'm not sure this is the right term, since in traditional domain swapping, a domain from one subunit would replace a similar domain from another subunit. In this case, the "domain swapped" PD takes up a unique position not otherwise occupied by a similar domain from another subunit. I would rather refer to it by the more general term "conformational switching." This terminological quibble aside, it is a fascinating phenomenon that also provides some insights into the possible assembly pathway of this shell (and by inference, T=13 shells in other viruses).

We agree that in traditional domain swapping, a domain from one subunit replaces a

similar domain from another subunit (we use the term as defined by Bennett et al. (1995)). As explained in the text, this is indeed what happens in our structure: the PD in an open-conformation replaces / occupies the space that would be otherwise taken up by the PD domain of another subunit in a closed conformation. (The PD does not take up a unique position as the reviewer suggests.) In this light, we feel the term domain-swapping is appropriate.

The authors' musings on the evolutionary implications of their structure are interesting. It is well known, of course, that conformational switching is important to build large icosahedral shells, but I think they are overselling the generality of the process. As they point out, even other T=13 dsRNA viruses do not appear to have a similar switch.

We are thankful for this comment. We have modified the text in the Discussion as follows: "our results allow us to propose a hypothetical model how closed viral protein shells may further evolve from such proteins" to further emphasise the fact that this is a model. Further studies will be needed to address its generality and we have added a comment to this end at the end of Discussion: "Further studies are required to assess to what extent domain swapping may have played a role in other viral systems."

Otherwise, the paper is well written and figures look nice. However, it is hard to figure out from Fig 2 which subunits correspond to Q, R, S and T and exactly how the trimers are organized in the context of the icosahedral asymmetric unit. A schematic diagram would be helpful.

We agree that the organization is quite complex. To help the reader to figure out which subunit is which, we have used a consistent colouring between the figures and we note in the legend of Figure 2: "The P8 trimers are coloured in yellow (P8_Q), green (P8_R), gold (P8_S), and brown (P8_T)".

Line 132: leavo → laevo.

Corrected.